# GC-MS, GC-IMS, and E-Nose Analysis of Volatile Aroma Compounds in Wet-Marinated Fermented Golden Pomfret Prepared Using Different Cooking Methods

**DOI:** 10.3390/foods13030390

**Published:** 2024-01-25

**Authors:** Qiuhan Chen, Xuebo Yang, Pengzhi Hong, Meijiao Liu, Zhuyi Li, Chunxia Zhou, Saiyi Zhong, Shouchun Liu

**Affiliations:** 1College of Food Science and Technology, Guangdong Ocean University, Zhanjiang 524088, China; chenqiuh1120@126.com (Q.C.); 18193642272@163.com (X.Y.); hongpz@gdou.edu.cn (P.H.); lmj2504@163.com (M.L.); s2416001752@163.com (Z.L.); chunxia.zhou@163.com (C.Z.); zhongsy@gdou.edu.cn (S.Z.); 2Guangdong Provincial Key Laboratory of Aquatic Products Processing and Safety, Zhanjiang 524088, China; 3Guangdong Provincial Engineering Technology Research Center of Marine Food, Zhanjiang 524088, China; 4Southern Marine Science and Engineering Guangdong Laboratory (Zhanjiang), Zhanjiang 524088, China; 5Guangdong Provincial Modern Agricultural Science and Technology Innovation Center, Zhanjiang 524088, China

**Keywords:** GC-MS, GC-IMS, E-nose, cooking methods, flavor

## Abstract

The cooking method is extremely important for the production of low-salt, wet-marinated, fermented golden pomfret because it strongly influences its flavor components and organoleptic quality. There are also significant differences in flavor preferences in different populations. The present study analyzed differences in the aroma characteristics of wet-marinated fermented golden pomfret after boiling, steaming, microwaving, air-frying, and baking using a combination of an electronic nose, GC-IMS, and SPME-GC-MS. Electronic nose PCA showed that the flavors of the boiled (A), steamed (B), and microwaved (C) treatment groups were similar, and the flavors of the baking (D) and air-frying (E) groups were similar. A total of 72 flavor compounds were detected in the GC-IMS analysis, and the comparative analysis of the cooked wet-marinated and fermented golden pomfret yielded a greater abundance of flavor compounds. SPME-GC-MS analysis detected 108 flavor compounds, and the results were similar for baking and air-frying. Twelve key flavor substances, including hexanal, isovaleraldehyde, and (E)-2-dodecenal, were identified by orthogonal partial least-squares discriminant analysis (OPLS-DA) and VIP analysis. These results showed that the cooking method could be a key factor in the flavor distribution of wet-marinated fermented golden pomfret, and consumers can choose the appropriate cooking method accordingly. The results can provide theoretical guidance for the more effective processing of fish products and the development of subsequent food products.

## 1. Introduction

Golden pomfret (*Trachinotus ovatus*) lives in tropical and temperate oceans, and it is primarily distributed in China’s Bohai Sea, the East China Sea, the South China Sea, and other waters. Its meat is delicious, nutritious, and rich in a large number of essential amino acids and unsaturated fatty acids that are needed by the human body [1]. Curing and fermentation are commonly used to prolong the preservation time of golden pomfret in the southern coastal areas of China [2]. Previous studies showed that fermentation imparted a unique fermented flavor to golden pomfret. The cooking treatment of this fish improves its safety for consumption and imparts a good flavor. GC-MS, GC-IMS, and electronic nose methods are widely used for food flavor analysis [3]. GC-MS qualitatively and quantitatively detects volatile compounds in food, but it cannot provide sensory information. GC-IMS visualizes the data, but the two-dimensional qualitative NIST Gas-Phase Retention Index Database and the IMS Migration Time Database of G.A.S. are not yet sufficiently complete in the current software. An electronic nose accurately and rapidly discriminates odors in different samples, but it cannot quantitatively detect odors [4]. These three techniques complement and validate each other to provide more comprehensive, reliable, and scientific information about food odors. GC-MS combined with an electronic nose can characterize and analyze the volatiles produced by different types of wood chips for the production of smoked bacon [5]. However, studies on comprehensive evaluations of flavor characterization and volatile compounds in fermented golden pomfret using the e-nose technique, GC-IMS, and GC-MS are rare.

Current research has focused on the effect of different cooking methods on nutritional, quality, and taste aspects [6,7]. Alexi [8] examined the changes in the nutritional and organoleptic qualities of white tigerfish and gilthead snapper fillets prepared using different cooking methods (steaming, oven cooking, and deep-frying). The results showed that the organoleptic characteristics of the samples subjected to steaming and oven cooking were similar, but deep-frying resulted in unfavorable increases in the n-6/n-3 and saturated fatty acid/polyunsaturated fatty acid (SFA/PUFA) ratios of the snapper [9]. Gladyshev [10] investigated the effects of different cooking methods (deep-frying, boiling, and baking) on essential polyunsaturated fatty acids in the muscle tissue of salmon and found that high levels of natural antioxidants protected against 20:5ω3 (EPA) and 22:6ω3 (DHA) during cooking. Tian Xiong [7] investigated and evaluated the effects of different cooking methods (steaming, deep-frying, microwaving, and baking) on several indicators, such as color, texture, steaming loss, nutrients, and volatile flavoring substances, in golden pomfret fillets to provide practical options for people. There are fewer reports on the effect of cooking methods on processed golden pomfret products, and no statistical analysis of results by combining multiple indicators. Therefore, it is necessary to compare the effects of different cooking methods on golden pomfret products.

The present study used an electronic nose, SPME-GC-MS, and GC-IMS to analyze the variability of flavor substances in wet-marinated fermented golden pomfret fillets processed by different cooking methods (steaming, baking, air-frying, microwaving, and poaching). Orthogonal partial least-squares discriminant analysis (OPLS-DA) was used to construct a model to identify the key flavor substances resulting from the different methods based on the variable projected importance factor (VIP). The characteristic flavor substances of pomfret can provide theoretical guidance for the more effective processing of fish products and the development of subsequent food products.

## 2. Materials and Methods

### 2.1. Materials

Golden pomfret (*Trachinotus ovatus*), with an average weight of 500.0 ± 10.0 g, was purchased from Huguang Market (Zhanjiang, Guangdong, China) within 1 h in a foam box with crushed ice and immediately stored at −40 °C for future analysis.

The mixed standard of n-alkanes (C_5_~C_32_) and 2,4,6-trimethylpyridine (chromatographic purity) were purchased from Shanghai Macklin Biochemi and Shanghai Amperexperiment Technology Co. (Shanghai, China), respectively. The other reagents and chemicals used were obtained from Sinopharm Chemical Reagent Co., Ltd. (Shanghai, China).

### 2.2. Preparation of the Sample

Samples (CK) of fresh golden pomfret were obtained by curing (salt concentration of 10%, shrimp water mass ratio of 1:3) and fermentation (fermentation conditions: 1% fermentation agent, fermentation for 24 h at 28 °C; fermentation agents: *Exiguobacterium profundum, Staphylococcus sciuri*, and *Staphylococcus gallinarum*).

The samples were prepared as follows:(1)A: Samples were boiled in water for 5 min, removed, cooled, and prepared for use.(2)B: Samples were steamed at 100 °C for 10 min, cooled, and prepared for use.(3)C: Samples were heated under 500 W microwave conditions for 5 min, cooled, and prepared for use.(4)D: Samples were baked in a 220 °C oven for 10 min, cooled, and prepared for use.(5)E: Samples were fried in an air fryer at 190 °C for 15 min, cooled, and set aside.

### 2.3. E-Nose Analysis

An electronic nose uses different sensors to detect the complex composition of a gas, and response priorities vary for different sensors. Thus, data processing methods are applied to identify a variety of odors and to analyze and evaluate the quality of the smell [11]. The experimental methodology was based on the methods of Tian P. and Siqueira A.F., with minor modifications [11,12]. The overall aroma profile of the samples was detected using a PEN3 electronic nose system (AirSense Analytics GmbH, Schwerin, Germany). A total of 5.0 g of the sample was weighed in a 40 mL headspace vial and sealed with a silicone stopper. The samples were equilibrated at 40 °C for 30 min and measured. The E-nose conducted measurements for 100 s and was cleaned for 120 s. The headspace vial was pumped at a constant rate of 400 mL/min into the sensor array (Table 1). All measurements were repeated three times.

### 2.4. GC-IMS Analysis

GC-IMS (Flavour Spec^®^, GAS, Dortmund, Germany) was used to analyze the volatile fingerprints of the samples. Following Xiaoshan Z.’s method [13], 2 g of the sample was weighed in a headspace vial and incubated at 60 °C for 10 min at an incubation speed of 500 r/min. Five hundred microliters of the headspace sample was injected into the headspace autosampler at 80 °C in splitless mode.

The following chromatographic conditions were used: the column temperature was 60 °C; the carrier gas was N_2_ (purity ≥ 99.999%); and the carrier gas flow rate was held for 2 min at 2.0 mL/min, linearly increased to 100 mL/min within 22 min, and held for 5 min.

IMS conditions: β-rays were used as the radiation source, and positive ions were used as the ionization mode in a drift tube (5.3 cm) operated at a constant temperature and flow rate of 45 °C and 150 mL/min, respectively.

### 2.5. SPME-GC-MS Analysis

Precisely 5.00 g of the treated sample was weighed in a 40 mL headspace vial, and then 2 μL of 2,4,6-trimethylpyridine standard solution was added. The vial was sealed using a cap. A solid-phase microextraction (SPME) needle (DVB/CAR/PDMS, 1 cm, 50/30 µm; Supelco, Bellefonte, PA, USA) was inserted into the vial containing the sample in the headspace. The sampling process was performed for 40 min in a water bath operating at a constant temperature of 65 °C. The needle was subsequently dispatched to the GC injection port for thermal desorption, which was performed at 250 °C for 5 min. The instrument was activated to retrieve the detection data [14]. A TQ8050NX gas chromatograph and mass spectrometer (Shimadzu, Kyoto, Japan) equipped with an InertCap^®^ Pure-WAX quartz capillary column (30 m × 0.25 mm, 0.25 μm) was used for GC-MS analysis. The carrier gas He (99.999% purity) was added at a flow rate of 1.0 mL/min. The flow rate was also maintained at 1.0 mL/min. The column was heated initially to 40 °C for 3 min, gradually increased to 100 °C at 4 °C/min for 2 min, and finally increased to 230 °C at 8 °C/min for 5 min. The electron ionization energy was 70 eV, the interface temperature was maintained at 250 °C, and the temperature of the ion source was 230 °C. The mass scanning range was between 33 and 550 *m*/*z*, and the acquisition mode was Q3 [15]. Notably, *m*/*z* represents the mass-to-charge ratio, which is a commonly used parameter in mass spectrometry.

### 2.6. Qualitative and Quantitative Analysis of Volatile Components

Volatile compounds may be characterized by the comparison of their mass spectra (MS) and retention indices (RIs, determined from n-alkanes C_5_–C_32_) with information from the National Institute of Standards and Technology (NIST) database. The magnitude of the odor activity value (OAV) determines the contribution of the volatile flavoring substance to the overall flavor [16]. Compounds with an OAV ≥ 1 were considered aromatically active compounds (AACs) with a significant effect on the aroma profile of the sample. The calculation method was as follows:(1)$$RI=100\times \left(\frac{t_{x}-t_{n}}{t_{n+1}+t_{n}}+n\right)$$
(2)$$OAV=\frac{C_{i}}{T}$$
where (1) *t_x_, t_n_*, and *t_n+_*_1_ represent the retention times of each volatile compound, n-carbon-atom n-alkane, and n+1-carbon-atom n-alkane, respectively (*t_n_* < *t_x_* < *t*_*n*+__1_); (2) Ci indicates the concentration of compound i (μg/kg); and T denotes the organoleptic odor threshold for the compound in water.

### 2.7. Data Analysis

Three replications of each set of experiments were performed, and the results are expressed as the means ± standard deviation. The data were analyzed and collated using the analysis software VOCal accompanying GC-IMS and GC×IMS Library Search(Flavour Spec^®^, GAS, Dortmund, Germany). The data were analyzed for significance using SPSS Statistics 26 (IBM, Armonk, NY, USA). A *p* value < 0.05 indicated a significant difference. OPLS-DA was performed using SIMCA-P 14.1 (Umetrics, Umea, Sweden), and the results were plotted using Origin 2019 (Origin Lab, Inc., Umea, Sweden).

## 3. Results

### 3.1. E-Nose Analysis

Electronic nose systems play an important role in the objective discrimination of volatile organic compounds (VOCs), and these systems are fast, simple, and reproducible [17]. The present study first analyzed different samples using an electronic nose, as shown in Figure 1A. The highest sensor response was obtained for W1W, followed by W5S, W1S, and W2S, which indicated that these samples contained high levels of sulfides, nitrogen oxides, alcohols, aldehydes, ketones, and methyl-containing compounds. The flavor fingerprints of B, A, and C were similar, and those of E and D were also similar. As shown in Figure 1B, sensor1 (W6S) contributed the most to the first principal component, sensor7 (W1W) contributed the most to the second principal component, and sensor2 (W5S) had a strong effect on both.

Principal component analysis (PCA) is an algorithm that assesses the overall differences between samples by extracting the principal components (PCs) of the data for dimensionality reduction. As shown in Figure 1C, PC1 and PC2 accounted for 88.78% and 9.66% of the variance, respectively, which indicated that these two components effectively explained the total variance. The PCA algorithm clearly separated the six groups into three distinct parts. Based on the intensity of the response of the 10 sensors to specific characteristic gases, the main characteristic gas of each sample was deduced. The loading analysis could determine the contribution of different sensors to the principal components and the ability of the sensors to distinguish between samples.

### 3.2. GC-IMS Analysis

GC-IMS analyzes the characteristic fingerprints of volatiles from different samples [18]. In the 2D spectra, the red vertical line with a migration time of 1.0 ms indicates the reactive ion peak (RIP) [19]. To compare different samples, CK spectra were selected as a reference in Figure 2a with the same concentration in white. Blue indicates a lower concentration, and red indicates a higher concentration. The number of peaks and signal intensities of the volatile organic compounds (VOCs) produced by different cooking methods of low-salt wet-marinated fermented golden pompano varied significantly. The fingerprints of the different treated samples were compared (Figure 2b). Each column represents a compound, and the color indicates the content of volatile compounds. The results showed that different cooking methods significantly affected the flavor of golden pomfret. Volatile compounds, including 2-hexanone, ethyl sulfide, and 2-methyl-2-pentenal, were found in all six samples and may be considered shared volatile flavor substances and the relative amounts of the substances in the red boxes are relatively similar. Figure 2b visually compares the differences in VOCs between samples. According to Figure 2b and Appendix A, a total of 72 volatiles were detected, including 17 alcohols, 10 aldehydes, 11 ketones, 11 esters, 6 acids, 4 ethers, 3 olefins, and 5 other volatiles.

Peak volume normalization was used to further summarize and compare the VOC content in different samples. As shown in Appendix A, the contents of aldehydes were 20.61% (A), 17.32% (E), 17.12% (D), 16.40% (CK), 11.42% (C), and 8.23% (B). Aldehydes, which are primarily generated by the oxidation and degradation of fatty acids, have a lower threshold than other compounds and have a greater impact on the overall flavor of fish samples even at low concentrations [20]. The highest levels found were of 3-methylbutyraldehyde (apple). Group B had significantly lower levels than the other cooking groups, and the lowest (*p* < 0.05) total aldehyde levels. High concentrations of aldehydes produce an unpleasant rancid odor [21]. The ketone content ranged from 16.25 to 27.11%, and the ketone compounds included acetoin (cream), 2-pentanone (fruit), and acetone (butter). The CK group had a significantly higher content than the other groups (*p* < 0.05). Although the ketone concentration threshold is low, it contributes positively to the flavor of fish samples [22]. The content of alcohols was 17.05–20.45% and included 3-methyl-3-buten-1-ol, 1-pentanol, ethanethiol, 3-methyl-2-butanol, 2-propanethiol, and 2-hexanol. Alcohols are generally produced from fatty acids via the decomposition of hydroperoxides catalyzed by lipid oxidases or via the reduction of carbonyl compounds. The alcohol content was approximately 20%, with no significant differences between groups [7]. Esters impart a fruity taste to meat products and have an important influence on flavor [23]. The content of esters was 10.86–15.59% and included propyl acetate, ethyl acrylate, propyl propionate, and other esters. Group D had a significantly greater content of phenolic compounds than the other groups (*p* < 0). Acids are primarily produced by the oxidation of fatty acid triglycerides or the microbial fermentation of amino acids [24]. The acid content was relatively low, but low-molecular-weight volatile acids contribute to the overall characteristic flavor of fish flesh. In conclusion, different cooking methods significantly affected the flavor of low-salt wet-marinated fermented golden pomfret fish.

### 3.3. GC-MS Analysis

The volatile organic compounds (VOCs) formed by different cooking methods in low-salt wet-salt fermented golden pomfret were analyzed qualitatively and quantitatively using SPME-GC-MS. A total of 108 volatile compounds were identified in six samples, including 24 aldehydes, 11 ketones, 10 alkanes, 10 alcohols, 15 olefins, 10 esters, 8 acids, and 20 other volatiles (Appendix A, Figure 3b). Among the 108 volatile compounds, there were 54 in CK, 49 in B, 51 in D, 41 in A, 51 in E, and 53 in C. There were differences in the types and contents of volatile compounds in the six groups of samples, with higher contents of aldehydes, alcohols, olefins, and others. The contents of aldehydes in the D group were much greater than in the other groups. The contents of aldehydes, acids, and aromatic compounds in the other groups were lower than in the CK group, which suggested that the samples were endowed with richer odors due to the production of more volatile compounds after cooking (Appendix A, Figure 3a).

Aldehydes have a low odor threshold and play a major role in the fishy taste of aquatic products. Related studies have shown that lipid oxidation and Strecker degradation in the Meladic reaction are the two main primary pathways for the production of aldehydes [25]. The total contents of aldehydes in the six groups were 38.77 μg/kg (CK), 55.18 μg/kg (B), 52.54 μg/kg (D), 51.26 μg/kg (A), 56.58 μg/kg (E), and 42.42 μg/kg (C). Nonanal and hexanal were the oxidation products of linolenic and linoleic acids, respectively, and primarily provided grassy and barbecue aromas, respectively, which contributed significantly to the flavor of the cooked golden pomfret [26]. Unsaturated aldehydes, such as (2E)-octen-1-al, (Z)-2-decenal, and (2E)-2,4-decadienal, are secondary oxidation products that are produced during the heating and oxidation of polyunsaturated fatty acids and manifest as fruity aromas.

Ketones are primarily produced via the thermal oxidative decomposition of saturated fatty acids, keto-enol inter-conjugated isomerization of hydroperoxides, further oxidation and decomposition of hydrocarbons, intramolecular electron rearrangements of peroxides in unsaturated fatty acids, and thermal oxidative degradation of amino acids [27]. The ketone content was significantly greater in the CK and C groups than in the other groups. The ketone concentrations in the six groups were 31.73 μg/kg (CK), 10.82 μg/kg (B), 19.26 μg/kg (D), 7.22 μg/kg (A), 6.81 μg/kg (E), and 35.43 μg/kg (C) and included acetoin (butter), 2-heptanone (bananas), 2-nonanone (fruits), and 2,3-octanedione (nuts) [28]. Alcohols are primarily produced via the oxidative breakdown of lipids. The higher thresholds for saturated alcohols contribute little to flavor, but the lower thresholds for unsaturated alcohols assist in flavor formation [29]. 1-Octen-3-ol is produced via the oxidative decomposition of arachidonic acid, and it is a typical flavoring substance for aquatic products that exhibits mushroom and earthy aromas. Phenylethanol is a common characteristic flavoring substance for fermented products with a rose aroma, and it assists in the formation of product flavor [18,30].

Hydrocarbons are primarily produced from the cleavage of fatty acid alkoxylates, but their higher threshold contributes less to the overall flavor. Some compounds, such as lauric acid, D-limonene, pinene, and limonene, contribute fruity, grassy, and lemony aromas to the flavor of the product [31]. Esters are generally produced from the esterification of acids and alcohols. Esters produced from short-chain acids have a fruity aroma, and esters produced from long-chain acids have a slightly greasy flavor. The higher contents of methyl 2-hydroxy-4-methylpentanoate and diisobutyl 2,2,4-trimethyl-1,3-pentanediol diisobutyrate confer a fruity aroma after cooking [32]. Small amounts of certain acids, such as caprylic acid, nonanoic acid, myristic acid, and n-pentadecanoic acid, have a fruity, coconutty, and waxy flavor, respectively, and were also detected [23].

The content of volatile flavor substances did not effectively indicate a key role in the overall flavor of the samples. Therefore, the samples were analyzed in the context of their own flavor thresholds. There were 15 key flavor substances with 0 < OAV < 1 in the different treatment groups (Table 2) and 11 key flavor substances with OAV ≥ 1, including hexanal, isovaleraldehyde, nonanal, (E)-2-duodenal, (E)-2-nonrenal, 2-octenal, decanal, acetoin, 2-nonanone, 1-octen-3-ol, and D-limonene. Nonanal, (E)-2-nonrenal, 1-octen-3-ol, D-limonene, and Estragole were the key compounds shared by the B, A, D, C, and E groups and contributed significantly to the formation of the flavor of cooked golden pomfret. Together, these compounds formed the flavor profile of golden pomfret, which was dominated by the aroma of oil and fat, with the aroma of green grass and a fruity flavor arising after cooking. The contents of the key flavor compounds in the six treatment groups were significantly different. Nonanal and 1-octen-3-ol were most prominent in the B and A groups and gave the low-salt wet-marinated golden pomfret a bland greasy flavor after high-temperature steaming. The key flavor compounds of hexanal, nonanal, and 1-octen-3-ol in the D and E groups had a significant effect on the overall flavor and gave the low-salt wet-marinated golden pomfret a mushroom and fruity aroma after high-temperature roasting and deep-frying. Hexanal, nonanal, and 1-octen-3-ol were also found in the C group. The OAVs of nonanal, acetoin, and 1-octen-3-ol were higher in group CK, and the golden pomfret after microwave cooking showed a mixed aroma of fat and fruit. The OAVs of nonanal, 2-octenal, acetoin, and 2-nonanone were lower in group CK, which indicated that the flavor was more muted than the other groups. In conclusion, the differences in the contents of key volatile flavor substances were an important reason for the differences in odor composition. A key cause of these differences was that the flavor profile of golden pomfret is richer after cooking. The interactions between the key flavor compounds also contributed significantly to the overall flavor of the samples.

Overall, different cooking methods had significant effects on the formation of certain volatile flavor compounds in the fish. Steaming treatments produced higher aldehyde contents and more unpleasant odors. Baking and air-frying treatments promoted the formation of more aggressive odors in the fish, which was likely due to the combination of high temperatures, protein denaturation, lipid oxidation, and the Maillard reaction.

#### OPLS-DA of Odor-Active Compounds from Different Cooking Methods

OPLS-DA is an analytical method for visualizing data and quantifying the degree of variation between samples using correlations between data [34]. The relative concentrations of substances with OAV > 1 in Table 2 and Appendix A were selected as the Y variables for the OPLS-DA modeling design. The explanatory power of the model for the X and Y matrices was expressed as R^2^X and R^2^Y, respectively, and the predictive power of the model was expressed as Q^2^, with an R^2^ and Q^2^ closer to 1.0 indicating a better fit for the model. Figure 4A shows that R^2^Y = 0.997 and Q^2^ = 0.994, which indicates that the model had good explanatory and predictive ability [35,36]. As shown in the figure, samples treated with different cooking methods were well separated. Y was located in the first quadrant, E was located in the second quadrant, D and A were located in the third quadrant, and C was located in the fourth quadrant. B was distributed in the second and third quadrants. The proximity of groups B and A suggests that the flavor types were similar.

The reliability of OPLS-DA was tested by performing 200 cross-replacement tests on the model, and the results are shown in Figure 4B. The horizontal coordinates in the graph are the retention of the samples, and the points at 1.0 are the R^2^ and Q^2^ of the original model. After validation, R^2^ (0.101) and Q^2^ (−0.735) were smaller than the retention value of 1.0, and the intercept of the model’s Q^2^ regression line with the horizontal coordinate was negative, which indicated that the model was free of overfitting and was stable and reliable.

The variable projection importance factor (VIP) is commonly used for key variable analysis in OPLS-DA models. A VIP greater than 1 indicates a greater contribution [37]. The VIP values for each key component are shown in Figure 4C. Components with VIP values greater than 1 were hexanal, isovaleraldehyde, (E)-2-dodecenal, (E)-2-nonenal, (E)-2-dodecenal, (E)-2-octenal, (E)-2-decenal, decanal, β-Myrcene, Limonene, and terpilene, which were identified as characteristic odor substances in combination with the OAVs and odor descriptions of key odor-activating substances.

## 4. Conclusions

The present study analyzed differences in the aroma characteristics of wet-marinated fermented golden pomfret after boiling, steaming, microwaving, air-frying, and baking using a combination of an electronic nose, GC-IMS, and SPME-GC-MS. The electronic nose differentiated the samples prepared by different cooking methods. A total of 72 flavor substances were detected using GC-IMS analysis, and wet-marinated fermented golden pomfret produced a richer range of flavor substances after comparative analysis. A total of 108 flavor substances were detected using SPME-GC-MS analysis, and the key flavor substances in different treatment groups were identified using OPLA-DA and VIP analyses. In conclusion, the cooking method is a key factor affecting the flavor distribution of wet-marinated fermented golden pomfret, and the method of cooking chosen by consumers may be used as a reference. At the same time, this study may provide a theoretical basis for future research and development regarding other golden pomfret products.

## Figures and Tables

**Figure 1 foods-13-00390-f001:**
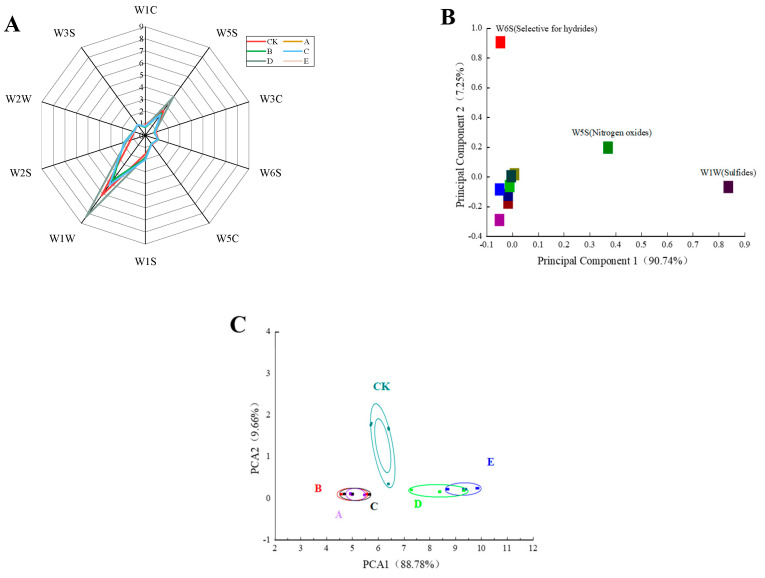
Electronic nose radar map of samples with different cooking methods (**A**), loading analysis of samples with different cooking methods (**B**), PCA two-dimensional map of samples with different cooking methods according to an electronic nose (**C**). CK: uncooked treated samples, A: boiled samples, B: steamed samples C: microwaved samples D: baked samples, E: air fried samples.

**Figure 2 foods-13-00390-f002:**
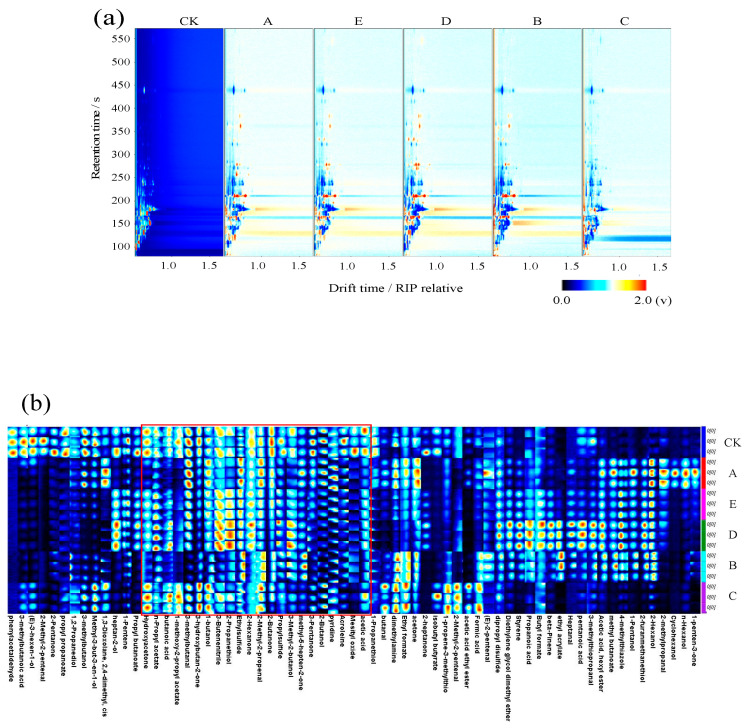
Two-dimensional topographic plot (**a**) and fingerprint spectra (**b**) of the different cooking methods used for GC-IMS. CK: uncooked treated samples, A: boiled samples, B: steamed samples C: microwaved samples D: baked samples, E: air fried samples.

**Figure 3 foods-13-00390-f003:**
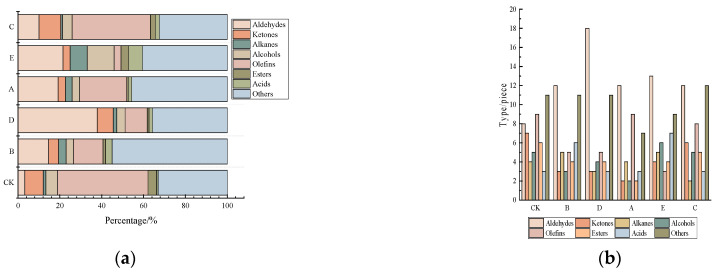
Contents (**a**) and types (**b**) of volatile substances detected using GC-MS for different cooking methods. CK: uncooked treated samples, A: boiled samples, B: steamed samples C: microwaved samples D: baked samples, E: air fried samples.

**Figure 4 foods-13-00390-f004:**
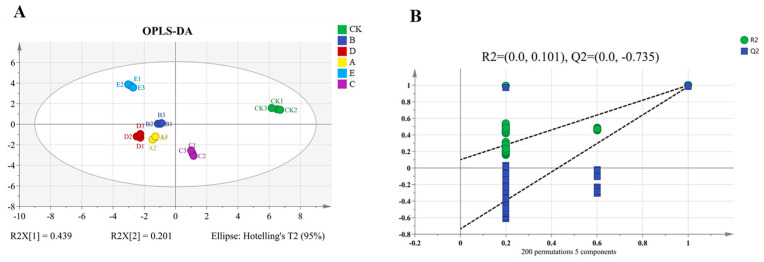
OPLS-DA score (**A**), cross-plot of 200 permutation tests (**B**), and VIP scores (**C**) for different cooking methods.

**Table 1 foods-13-00390-t001:** Sensors and corresponding representative sensitive material types of the PEN3 electronic nose.

Array Number	Sensor Name	Description of Sensitivity	Sensitive Gas	Threshold (mL/m^3^)
1	W1C	Aromatic components and benzene	C_7_H_8_	10
2	W5S	Nitrogen oxides	NO_2_	1
3	W3C	Aromatic components with ammonia	C_6_H_6_	10
4	W6S	Selective for hydrides	H_2_	100
5	W5C	Aromatic components of short-chain alkanes	C_3_H_8_	1
6	W1S	Methyl groups	CH_4_	100
7	W1W	Sulfides	H_2_S	1
8	W2S	Alcohols, aldehydes, and ketones	CO	100
9	W2W	Aromatic components and organosulfides	H_2_S	1
10	W3S	Long-chain alkanes	CH_4_	100

**Table 2 foods-13-00390-t002:** Odor thresholds and aroma-active compounds in different samples.

NO	Compound	Threshold (μg/kg) [33]	Odorant Description	CK	A	B	C	D	E
A1	Hexanal	4.5	green onion flavor, green fruit flavor	-	2.99	2.75	-	9.82	2.45
A2	Isovaleric aldehyde	13	apple, peach	0.47	0.78	-	1.51	1.61	-
A3	Nonanal	1	green onion, green fruit flavor	1.99	9.12	12.10	4.43	13.96	11.52
A4	(2E)-Dodecenal	1.4	citrus, fat	-	-	1.35	-	0.05	-
A5	U-ecanal	10	oil, pungent, sweet	0.04	-	0.14	0.09	-	0.83
A6	(2E)-Nonenal	0.69	cucumber, fat, green	-	0.44	0.80	1.22	0.76	1.23
A7	(E)-2-Dodecenal	1.4	soap	-	-	0.24	-	-	0.11
A8	Oct-2-enal	0.3	green, nut, fat	2.25	-	1.49	1.42	-	-
A9	(2E)-2-Octenal	3	fat	-	0.15	-	-	0.28	0.13
A10	(2E)-2,4-Decadienal	2.3	citrus, chicken	-	0.10	-	-	0.18	0.79
A11	(E)-Decenal	3	soap	0.06	-	-	-	-	0.47
A12	Decanal	1	soap, orange peel, tallow	-	-	-	1.24	-	1.61
A13	Acetoin	8	butter, cream	1.11	0.60	0.80	3.41	-	0.34
A14	Heptan-2-one	9	pears	0.37	-	-	0.18	-	-
A15	2-Undecanone	7	waxy, fruity, creamy, fatty, orris, floral	0.41	-	-	-	-	-
A16	2-Nonanone	5	green, weedy, earthy, herbal	1.90	-	-	0.32	-	-
A17	1-Octen-3-ol	1	mushroom	-	6.90	5.55	7.02	7.17	3.80
A18	(E)-2-Octen-1-ol	40	mushroom	0.13	-	-	0.04	0.05	-
A19	β-Myrcene	13	herb, wood, spice	0.10	-	-	0.26	-	-
A20	Limonene	4	lemon, orange	0.68	-	-	-	-	-
A21	D-Limonene	60	fruit	-	0.56	0.48	1.71	0.40	0.09
A22	Terpilene	85	lemon, orange	0.09	0.04	-	0.12	-	-
A23	γ-Decalactone	2.6	peach, fat	0.71	-	-	0.18	-	-
A24	Estragole	35	licorice, anise	0.89	0.34	0.54	0.49	0.25	0.14
A25	2-pentyl-Furan	6	fruity, green	-	-	-	0.27	-	0.23

Note: CK: uncooked treated samples, A: boiled samples, B: steamed samples C: microwaved samples D: baked samples, E: air fried samples. odor descriptions were retrieved from https://www.thegoodscentscompany.com/search2.html, accessed on 18 July 2023.

## Data Availability

The original contributions presented in the study are included in the article/Appendix A, further inquiries can be directed to the corresponding author.

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
