# Peer review of "GC-MS, GC-IMS, and E-Nose Analysis of Volatile Aroma Compounds in Wet-Marinated Fermented Golden Pomfret Prepared Using Different Cooking Methods"

_foods, 2024, doi:10.3390/foods13030390_

Round 1

Reviewer 1 Report

Comments and Suggestions for Authors

line 54: the authors wrote: "GC- IMS visualizes the data, but it lacks a complete database" but what does it mean ? Please extensively explain the sentence with adequate references.

line 66: there are some abbreviations that were not disclosed in the text e.g. "SFA/PUFA", EPAs and DHAs, CK, HK, E(A) ?

line160-161: Specify the source of T data (maybe [33])

line 187: "soybean oil" ??

Figure 1(b) I suggest to plot also loadings to verify the most reactive sensors discriminating samples. See also: http://dx.doi.org/10.1016/j.jclepro.2016.05.148

eq. (3) OVA or OAV ?

line 285: "(E)-2-duodenal, (E)-2-nonrenal" are you sure of these names ? Maybe a table with structure of the main OAV>1 compounds should be useful

line 311 "Merad reaction" ??? maybe Maillard ?

Comments on the Quality of English Language

Some typing errors are present but english is almost good enough for publication. 

Author Response

Dear Reviewer:

Thank you for the reviewer's comments concerning our manu entitled "GC-MS, GC-IMS and E-Nose Analysis of Volatile Aroma Compounds in Wet-Marinated Fermented Golden Pomfret Using Different Cooking Methods". Those comments are all valuable and very helpful for revising and improving our paper, as well as the important guiding significance to our researches. We have studied comments carefully and have made correction which we hope meet with approval. Revised portion are marked in red in the paper. The responds to the reviewer's comments are as flowing:

line 54: the authors wrote: "GC- IMS visualizes the data, but it lacks a complete database" but what does it mean ? Please extensively explain the sentence with adequate references.

Response: The current database used by GC-IMS is not fully searchable for all compounds, and we use it for aroma detection mainly for its data visualization advantages.

See also: Detection of volatile compounds in re-stewed chicken by GC-IMS;

line 66: there are some abbreviations that were not disclosed in the text e.g. "SFA/PUFA", EPAs and DHAs, CK, HK, E(A) ?

Response: We have explained these abbreviations in the original article.

line160-161: Specify the source of T data (maybe [33])

Response: Specific data sources are labeled in Table 4,so that's why it's not preceded by a source

line 187: "soybean oil" ??

Response: We have made correction according to the Reviewer's comments.

Figure 1(b) I suggest to plot also loadings to verify the most reactive sensors discriminating samples. See also: http://dx.doi.org/10.1016/j.jclepro.2016.05.148

  1. (3) OVA or OAV ?

Response: We have made correction according to the Reviewer's comments and added load charts to the original article.

line 285: "(E)-2-duodenal, (E)-2-nonrenal" are you sure of these names ? Maybe a table with structure of the main OAV>1 compounds should be useful

Response: The structures of the volatile flavor compounds are shown in the supplementary material to Table 4

line 311 "Merad reaction" ??? maybe Maillard ?

Response: We have made correction according to the Reviewer's comments.

Thank you very much for your comments and suggestions. For specific changes, please refer to the Appendix.

Reviewer 2 Report

Comments and Suggestions for Authors

GC-MS, GC-IMS and E-Nose Analysis of Volatile Aroma Com- 2 pounds in Wet-Marinated Fermented Golden Pomfret Using 3 Different Cooking Methods

Abstract

1.      The abstract is clear and concise. Whole abstract is written in the past tense why the last line of abstract (line # 39 and 40) is in present. “These results show that cooking method is a key factor in the flavor distribution of wet-marinated fermented golden pomfret, and consumers 40 can choose the appropriate cooking method accordingly”

2.      The abstract concludes with a statement on the importance of cooking methods, which is good. However, consider adding a brief sentence that highlights the practical implications of the findings for consumers or the food industry.

Introduction

3.      While the first paragraph mentions that GC-MS, GC-IMS, and electronic nose methods complement each other, it would be beneficial to explain why these specific techniques were chosen and how they contribute to the study's objectives.

4.      The passage effectively references previous research on cooking methods' effects on other fish species, providing a background for the study. However, it would be helpful to explain how the current study differs or adds to this existing knowledge.

Material Methodology                                

5.      Provide more details on the curing and fermentation conditions, such as the specific fermentation agent used and any additional parameters relevant to the preservation process

6.      Clarify the significance of using an electronic nose for aroma profile detection. Explain how the electronic nose system works, its sensitivity, and the relevance of the parameters measured

7.      Include a brief sentence or phrase that transition to the next part of the methodology or explains how the information provided fits into the overall analysis of volatile fingerprints

Conclusion

8.      Explore future research directions, such as investigating the sensory preferences of consumers to determine which flavor profiles are more appealing.

9.      Provide a clearer link between the findings and real-world applications, making the study more accessible and relevant to a broader audience.

Author Response

Dear Reviewers:

Thank you for the reviewer's comments concerning our menu entitled "GC-MS, GC-IMS and E-Nose Analysis of Volatile Aroma Compounds in Wet-Marinated Fermented Golden Pomfret Using Different Cooking Methods". Those comments are all valuable and very helpful for revising and improving our paper, as well as the important guiding significance to our research. We have studied comments carefully and have made correction which we hope to meet with approval. Revised portions are marked in red in the paper. The responds to the reviewer's comments are as flowing:

Abstract

1.The abstract is clear and concise. Whole abstract is written in the past tense why the last line of abstract (line # 39 and 40) is in present. “These results show that cooking method is a key factor in the flavor distribution of wet-marinated fermented golden pomfret, and consumers 40 can choose the appropriate cooking method accordingly”

2.The abstract concludes with a statement on the importance of cooking methods, which is good. However, consider adding a brief sentence that highlights the practical implications of the findings for consumers or the food industry.

Response: We have made correction according of grammar problems to the Reviewer's comments. It also expands on the implications of the findings for consumers and the food industry.

Introduction

3.While the first paragraph mentions that GC-MS, GC-IMS, and electronic nose methods complement each other, it would be beneficial to explain why these specific techniques were chosen and how they contribute to the study's objectives.

4.The passage effectively references previous research on cooking methods' effects on other fish species, providing a background for the study. However, it would be helpful to explain how the current study differs or adds to this existing knowledge.

Response: Our reasons for choosing a combined GC-MS, GC-IMS and E-nose assay and the advantages of applying it to the assay are supplemented in the text. Previous studies have compared the effects of different cooking methods on the quality, nutritional value, and flavor of samples. Comparison with the current study shows that newer, more efficient assays allow for a more diverse presentation of experimental results.

Material Methodology  

5.Provide more details on the curing and fermentation conditions, such as the specific fermentation agent used and any additional parameters relevant to the preservation process

6.Clarify the significance of using an electronic nose for aroma profile detection. Explain how the electronic nose system works, its sensitivity, and the relevance of the parameters measured.

7.Include a brief sentence or phrase that transition to the next part of the methodology or explains how the information provided fits into the overall analysis of volatile fingerprints

Response: We've accepted your suggestion and elaborated on some of the process parameters in the materials section of the article. In section 2.3, the principle of using the electronic nose and the significance of using the electronic nose is explained, and in section 3.1, a load diagram is added to supplement the data. We've also added transition sentences between paragraphs that continue from one to the next.

Conclusion

8.Explore future research directions, such as investigating the sensory preferences of consumers to determine which flavor profiles are more appealing.

9.Provide a clearer link between the findings and real-world applications, making the study more accessible and relevant to a broader audience.

Response: We have made correction according to the Reviewer's comments.

Thank you very much for your comments and suggestions. For specific changes, please refer to the Appendix.
